# Arsenic Removal by Advanced Electrocoagulation Processes: The Role of Oxidants Generated and Kinetic Modeling

**Micah Flor V. Montefalcon** [1], **Meliton R. Chiong III** [1], **Augustus C. Resurreccion** [2], **Sergi Garcia-Segura** [3] **and Joey D. Ocon** [1,*]

1. Laboratory of Electrochemical Engineering (LEE), Department of Chemical Engineering, College of Engineering, University of the Philippines Diliman, Quezon City 1101, Philippines; mvmontefalcon@up.edu.ph (M.F.V.M.); mrchiong@msep.upd.edu.ph (M.R.C.III)
2. Institute of Civil Engineering, College of Engineering, University of the Philippines Diliman, Quezon City 1101, Philippines; acresurreccion@up.edu.ph
3. Nanosystems Engineering Research Center for Nanotechnology-Enabled Water Treatment, School of Sustainable Engineering and the Built Environment, Arizona State University, Tempe, AZ 85287-3005, USA; Sergio.Garcia.Segura@asu.edu
* Correspondence: jdocon@up.edu.ph

**Abstract:** Arsenic (As) is a naturally occurring element in the environment that poses significant risks to human health. Several treatment technologies have been successfully used in the treatment of As-contaminated waters. However, limited literature has explored advanced electrocoagulation (EC) processes for As removal. The present study evaluates the As removal performance of electrocoagulation, electrochemical peroxidation (ECP), and photo-assisted electrochemical peroxidation (PECP) technologies at circumneutral pH using electroactive iron electrodes. The influence of As speciation and the role of oxidants in As removal were investigated. We have identified the ECP process to be a promising alternative for the conventional EC with around 4-fold increase in arsenic removal capacity at a competitive cost of 0.0060 \$/$m^3$. Results also indicated that the rate of As(III) oxidation at the outset of electrochemical treatment dictates the extent of As removal. Both ECP and PECP processes reached greater than 96% As(III) conversion at 1 C/L and achieved 86% and 96% As removal at 5 C/L, respectively. Finally, the mechanism of As(III) oxidation was evaluated, and results showed that Fe(IV) is the intermediate oxidant generated in advanced EC processes, and the contribution of ●OH brought by UV irradiation is insignificant.

**Keywords:** arsenite and arsenate removal; electrochemical peroxidation; peroxicoagulation; Fenton-like processes; photo-assisted processes

## 1. Introduction

Arsenic (As) is one of the top ten pollutants associated with water quality violations. Arsenic contamination of groundwater has been reported from over 70 countries, affecting around 150 million people worldwide [1]. Approximately 110 million of those who are affected live in South and South-east Asia [2]. In the Philippines, a case of chronic arsenic poisoning was reported to the Department of Health in 2014 [3]. Upon data validation, it was found that 215 people had consulted with similar dermatological symptoms since 2010. Groundwater quality tests conducted showed arsenic concentrations greater than 10 μg/L, which is the maximum limit set by the Philippine National Standards for Drinking Water (PNSDW) for arsenic [4].

Arsenic is a ubiquitous element in the environment and is a constituent of more than 300 minerals [5]. Excessive amounts of arsenic in groundwater may be due to natural sources, such as weathering of arsenic-containing minerals. However, increasing concentrations have been associated with anthropogenic activities including uncontrolled industrial effluents release and application of organo-arsenical pesticides [6]. Chronic exposure to high levels of this heavy metal can cause significant health risks such as skin lesions, cancer, and complications on the respiratory, nervous, and reproductive systems [7]. Soluble inorganic ions arsenite [As(III)] and arsenate [As(V)] are the two most prevalent forms of arsenic found in natural water sources, with As(III) being more toxic, mobile, and soluble than As(V) [8]. Moreover, As(III) is hardly removable, and treatment strategies consist of pre-oxidation to As(V) to achieve complete As removal from the aqueous phase. Conventional technologies used to remove arsenic generally consist of two steps: (1) As(III) oxidation to As(V) using traditional oxidants such as chlorine [9] or permanganate [10]; followed by (2) As(V) physical removal via coagulation-flocculation, membrane technologies, adsorption, or ion exchange [11–13].

Electrocoagulation (EC), an emerging water treatment technology, has recently gained attention due to its simplicity in operation, minimum sludge production, and low capital, operating, and maintenance cost requirements [14]. One of its major advantages over conventional processes is that it can effectively oxidize As(III) to As(V) by direct charge transfer which enhances total arsenic removal [6,15,16]. In EC, a sacrificial anode, typically iron or aluminum, is utilized as a source of coagulating ions [14]. For the treatment of arsenic-contaminated wastewaters, however, iron electrode was found to increase arsenic removal [16]. Sacrificial iron electrodes are oxidized yielding Fe(II) ions via reaction, Equation (1):

$$Fe(0) \rightarrow Fe^{2+} + 2e^- \tag{1}$$

Reactive oxygen species in solution (e.g., $O_2$) can further oxidize Fe(II) to Fe(IV) by Equations (2) to (6):

$$Fe^{2+} + O_2 \rightarrow O_2^{\bullet-} + Fe^{3+} \tag{2}$$

$$Fe^{2+} + O_2^{\bullet-} + 2H^+ \rightarrow Fe^{3+} + H_2O_2 \tag{3}$$

$$Fe^{2+} + H_2O \rightleftharpoons Fe^{II}(OH)^+ + H^+ \tag{4}$$

$$Fe^{II}(OH)^+ + H_2O_2 \rightarrow (OH)Fe(H_2O_2)^+ \tag{5}$$

$$(OH)Fe(H_2O_2)^+ \rightarrow Fe^{IV}O^{2+} + H_2O + OH^- \tag{6}$$

Generated Fe(IV) is an oxidant species that may act as a redox mediator promoting As(III) oxidation to As(IV) (Equation (7)) while yielding Fe(III) that is known to improve coagulation performance due to the insolubility of Fe(III) hydroxylated species as well as its higher charge density. Furthermore, Fe(IV) can directly oxidize Fe(II) to Fe(III) via Equation (8).

$$Fe^{IV}O^{2+} + As(III) + H_2O \rightarrow As(IV) + Fe^{3+} + 2OH^- \tag{7}$$

$$Fe^{IV}O^{2+} + Fe^{2+} + H_2O \rightarrow 2Fe^{3+} + 2OH^- \tag{8}$$

The Fe(III) ions produced then forms hydrous ferric oxide (HFO) precipitates with high arsenic sorption affinity [17]. At the cathode, $H_2$ gas is formed, which aids in the flotation of coagulated particles and mass transfer in the solution (Equation (9)) [16].

$$2H^+ + 2e^- \rightarrow H_{2(g)} \tag{9}$$

The efficiency of the electrocoagulation process is influenced by a number of factors such as pH, current density, charge dosage rate, electrode material used, reactor design parameters, and concentration of other ions present [16,18,19].

Emerging advanced EC processes can improve arsenic removal but have been barely explored in literature for the removal of inorganic species [18]. These are emergent technologies that use the simultaneous generation of in situ •OH and other oxidants to improve pollutant removal [18]. One example of this is electrochemical peroxidation (ECP). The process is similar to that of EC, but here, $H_2O_2$ is externally added to the solution. The electrogenerated Fe(II) ions react with the hydrogen peroxide to produce Fe(III), which in turn yields hydrous ferric oxide (HFO) [20]. Depending on the solution's pH, •OH or Fe(IV) species can be formed as intermediates, which can oxidize both As(III) and Fe(II).

Another advanced EC process is the photo-assisted electrochemical peroxidation (PECP). Here, hydroxyl radicals are produced either by $H_2O_2$ photolysis (Equation (10)), Fenton's reaction (Equation (11)), or photoreduction of $Fe(OH)^{2+}$ (Equation (12)).

$$H_2O_2 \overset{h\nu}{\rightarrow} 2\bullet OH \tag{10}$$

$$Fe^{2+} + H_2O_2 \rightarrow Fe^{3+} + \bullet OH + OH^- \tag{11}$$

$$Fe(OH)^{2+} \overset{h\nu}{\rightarrow} Fe^{2+} + \bullet OH \tag{12}$$

UV irradiation also offers an advantage by improving the rate of the Fenton-type reactions thereby improving As removal. It is also used in disinfecting water with microorganisms as UV damages their nucleic acid leaving them unable to perform their vital functions [18]. This system can then provide arsenic removal and disinfection to ensure access to drinking water as a decentralized treatment of groundwater and surface water sources for developing regions.

Previous studies have investigated the oxidation of As(III) by UV/$H_2O_2$ [21–25], and the removal of arsenic using EC [6,15,26–30] and ECP [31,32]. Meanwhile, a limited number of papers have explored the PECP process mainly for the treatment of organics via •OH formation [33–37]. Elucidation of the role of electrogenerated oxidants on the removal of As(III) can allow identifying enhanced removal treatments, as well as understanding the fundamental mechanisms of the kinetic process. Therefore, modeling approaches can provide a guideline for future advances to overcome the arsenic pollution challenge that detrimentally affects developing communities. Evaluation of these emerging processes under realistic pollutant concentrations is required to assess potential applicability and technology competitiveness [38].

In this study, different EC and advanced EC technologies are benchmarked. The performance of these processes in terms of As(III) oxidation and removal at circumneutral pH was investigated. Specifically, the objectives of this work were: (i) to compare the % As(III) removal of EC, ECP, and PECP technologies, (ii) to examine the influence of arsenic speciation on arsenic removal, (iii) to investigate the role of oxidants in arsenic removal, and (iv) to evaluate the mechanism of As(III) oxidation in advanced EC processes.

## 2. Results and Discussion

### 2.1. Benchmarking Conventional Electrocoagulation with Advanced Technologies

The treatment of solutions containing realistic arsenic concentrations found in groundwater of 500 µg/L allowed us to compare the removal performance of EC, ECP, and PECP. Figure 1 illustrates the arsenic removal and the threshold objective defined by the WHO as the maximum contaminant level (MCL) for drinking waters of 10 µg/L [39]. Noteworthy differences were observed in terms of charge loading requirements to decrease arsenic concentration below MCL. Decreasing values of circulated charges of 27.6 C/L, 6.7 C/L, and 4.9 C/L were required for EC, ECP, and PECP, respectively. These results suggest higher removal performance of advanced electrocoagulation processes. This trend can be explained by the faster oxidation of As(III) to As(V) mediated by oxidants generated in the bulk, impacting the physical separation through adsorption/coagulation.

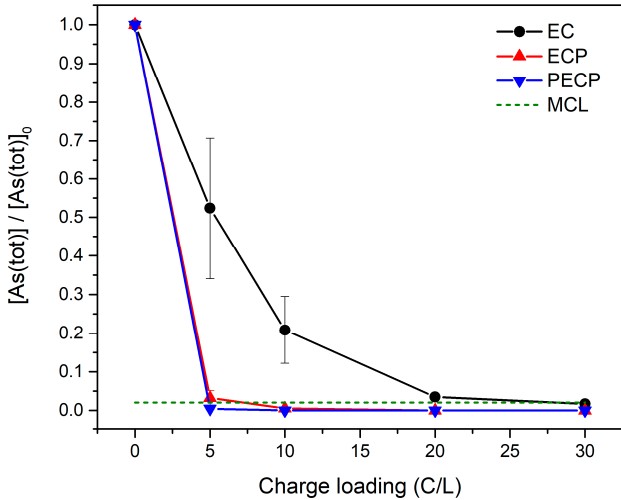

**Figure 1.** Removal of total arsenic from aqueous solutions containing ~500 μg/L of As(III) at pH 7 by different physical separation electrochemically driven processes at 1 C/L/min: (●) electrocoagulation, (▲) electrochemical peroxidation, (▼) photo-assisted electrochemical peroxidation. Dashed line indicates the maximum contaminant level of 10 μg/L recommended by the World Health Organization.

Even though there is ca. 6-fold reduction in operational time for the PECP process with respect to conventional EC, further analysis of techno-economical aspects provides a different perspective in terms of competitiveness of these electrochemically driven processes [40]. Table 1 compares the three electrocoagulation-based technologies in terms of removal capacity and cost per volume treated. It can be seen how the enhanced oxidation performance of advanced electrocoagulation technologies results in a considerably higher removal performance of As in comparison to conventional EC (57 μg/L) of ca. 4-fold for ECP (218 μg/L) and outstanding ca. 7-fold for PECP (384 μg/L). However, a higher operational cost is associated with the PECP process due to the higher energy requirement of UV lamps. This effect drastically increases energy requirements and detrimentally impacts cost reaching values of 0.249 $/m$^3$, which is 40 times higher than conventional EC. The cost evaluation suggests that PECP is not cost-competitive. The advanced ECP process shows competitive costs to EC of 0.0060 $/m$^3$ with improved performance. Note that ECP has an expected longer operational life of sacrificial electrodes (at least 4 times longer) due to the lower iron dosage requirement and the shorter treatment time. This is expected to have an impact on operational costs associated with labor and off-periods for electrode substitution. Moreover, it is to be noted that the longer treatment time required by EC does not only affect operational expenditures. Indeed, the required reactor design at larger scales may result in treatment units of higher physical footprint that will also increase capital costs. In this frame, the ECP process can be identified as a promising alternative to enhance arsenic removal from water.

**Table 1.** Treatment time, arsenic removal capacity, and operating cost required for electrocoagulation (EC), electrochemical peroxidation (ECP), and photo-assisted electrochemical peroxidation (PECP) processes to reach maximum contaminant level (MCL).

| Treatment | $t_{EC, WHO}$ | As Removal Capacity | | Operating Cost | | | |
|---|---|---|---|---|---|---|---|
| | min | μg As removed/Coulomb | μg As removed/mg Fe | Energy, $/m$^3$ | Electrode, $/m$^3$ | $H_2O_2$, $/m$^3$ | Total, $/m$^3$ |
| EC | 27.6 | 16.6 | 57.3 | 0.0014 | 0.0048 | - | 0.0062 |
| ECP | 6.7 | 63.0 | 217.8 | 0.0003 | 0.0012 | 0.0045 | 0.0060 |
| PECP | 4.9 | 111.1 | 384.0 | 0.2436 | 0.0009 | 0.0045 | 0.2490 |

*2.2. Arsenic Speciation during Electrochemical Treatment*

Arsenic speciation is a parameter that can highly affect treatment performance. Figure 2 shows the time-course of As(III), As(V), and As(tot) concentrations with charge loading during different

EC-based processes. Arsenic removal performance is directly associated with the speciation between As(III) and As(V), being As(V) easily removed by adsorption on Fe(III) coagula.

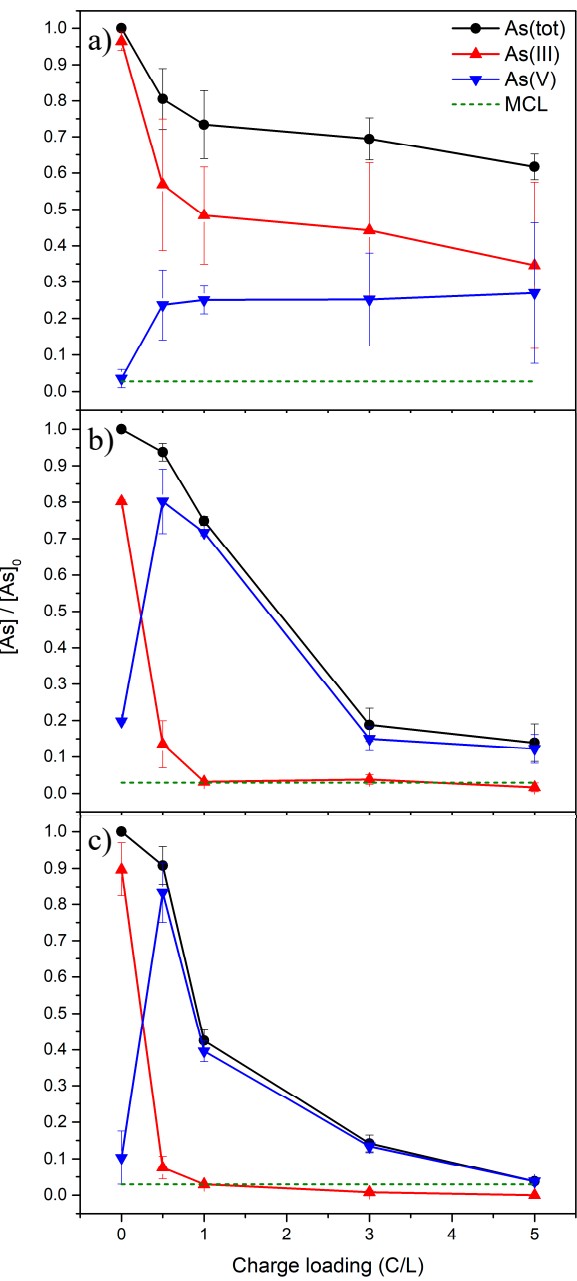

**Figure 2.** Speciation of arsenic during (**a**) electrocoagulation, (**b**) electrochemical peroxidation, (**c**) photo-assisted electrochemical peroxidation: (●) $As_{total}$, (▲) As(III), (▼) As(V). Dashed line indicates the maximum contaminant level of 10 μg/L recommended by the World Health Organization.

In EC, As(V) formation was initially fast with 40% As(III) converted to As(V) at 0.5 C/L. Afterward, slow As(III) oxidation was observed from 0.5 to 5 C/L; and only an additional 22% As(III) was converted. Arsenic removal was initially fast, with 19% As removed at 0.5 C/L. From 0.5 C/L onward, a slow reduction in As concentration was observed with the amount of As removed doubling at 5 C/L. It was also noted that on average, only 53% of As(V) formed is removed from the solution. This trend suggests that the amount of HFO formed was insufficient to remove the remaining As(V). Using this treatment, 38% of the initial total As was removed after the application of 5 C/L.

In contrast, As removal was initially slow in ECP and PECP processes with only 6% and 9% removal at 0.5 C/L, respectively. However, As(V) formation occurred rapidly at the start of both treatments reaching greater than 96% As(III) conversion at 1 C/L charge loading. This initially slow As removal indicates that the oxidants present in these advanced EC processes were consumed first in the As(III) oxidation reaction. Once almost all As(III) was oxidized, As(V) concentration decreased rapidly with charge loading as HFO was continuously formed. After the application of 5 C/L, the As removal achieved using ECP and PECP processes were 86% and 96%, respectively.

From Figure 2, it can be observed that the rate of As(III) oxidation at the outset of electrolysis provides information regarding the amount of available oxidant in the solution and the extent of As removal. Because As(V) formed is adsorbed rapidly, As(tot) concentration decreases dramatically provided that there are enough HFO precipitates available in the solution. As in the case of EC, the low As(III) oxidation rate at the beginning of the treatment suggests that there is an insufficient amount of oxidant present, which resulted in low As removal. On the other hand, the high As(III) oxidation rates at the start of ECP and PECP processes indicate a sufficient supply of oxidants, thereby achieving high As removal. Understanding of As(III) to As(V) oxidation, thus, provides a better insight into the mechanism of As removal.

### 2.3. Understanding the Role of Oxidants in Arsenic Removal

Arsenic removal performance is directly associated with the speciation between As(III) and As(V), being As(V) easily removed by adsorption on Fe(III) coagula. In this context, understanding how different processes influence the concentration of As(III) during treatment is essential. Figure 3 illustrates the As(III) concentration profile under different operation conditions. Conventional EC treatment oxidizes 65% of As(III) by direct charge transfer. A blank experiment conducted solely in the presence of $H_2O_2$ demonstrated the chemical oxidation of As(III) with a similar removal close to 50%. Whereas, the combination in advanced ECP and PECP suggests a synergistic interaction that accelerates As(III) oxidation. This trend can be explained by the generation of redox mediators such as •OH with a high standard reduction potential of $E° = 2.8$ V vs. SHE [41] or Fe(IV) with $E° = {\sim}2.0$ V vs. SHE [42]. Accordingly, As(III) oxidation yields were found to increase in the sequence: EC < $H_2O_2$ < ECP < PECP, with 42%, 51%, 83%, and 92% conversion at 0.5 C/L charge loading, respectively. These results agree with the trends observed in Figure 1 with higher removals attained for advanced EC processes.

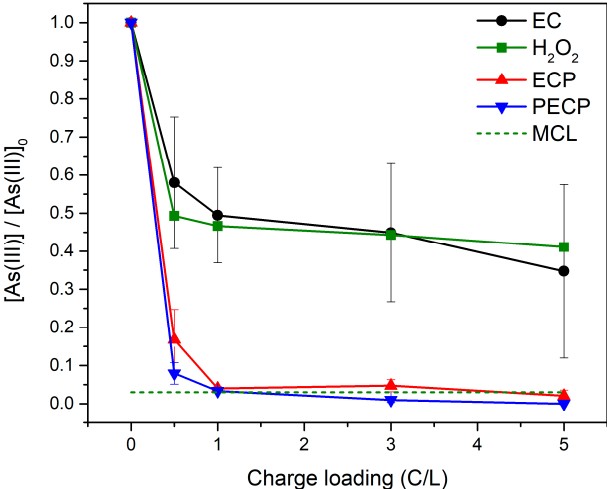

**Figure 3.** As(III) concentration profile using different treatment processes: (•) electrocoagulation, (■) oxidation using $H_2O_2$, (▲) electrochemical peroxidation, (▼) photo-assisted electrochemical peroxidation (initial As(III) = 331±21 µg/L; initial pH = 7; charge dosage rate = 1 C/L/min; $H_2O_2$ concentration = 5 mg/L).

Experiments in the presence of specific scavengers were conducted to identify the role of oxidants generated. Figure 4 shows the profile of As(III) in the presence of 2-propanol (14mM) as •OH scavenger. According to previous studies, 2-propanol and As(III) react similarly with •OH, with rate constants of $1.9 \times 10^9$ L mol$^{-1}$ s$^{-1}$ [43] and $8.5 \times 10^9$ L mol$^{-1}$ s$^{-1}$ [44], respectively. At the added concentration of 2-propanol, it would completely inhibit the oxidation of As(III). However, no significant difference in As(III) oxidation and removal was noted after the addition of 2-propanol in both ECP and PECP treatments (Figure 4). Thus, •OH can be disregarded as the oxidant involved in the fast oxidation of As(III) reported in Figure 3. This can be explained by the different chemistry of Fenton processes depending on the solution pH. Previous studies reported that the reaction between Fe(II) and $H_2O_2$ undergoes a mechanistic change at different pH. The yield of •OH is maximized under acidic conditions, whereas the production of Fe(IV) species becomes predominant at circumneutral and alkaline pH [45,46]. This hypothesis aligns with the observed trends in Figure 4. Therefore, it is suggested that the mechanistic oxidation of As(III) is mediated by Fe(IV) species, as summarized in the scheme shown in Figure 5.

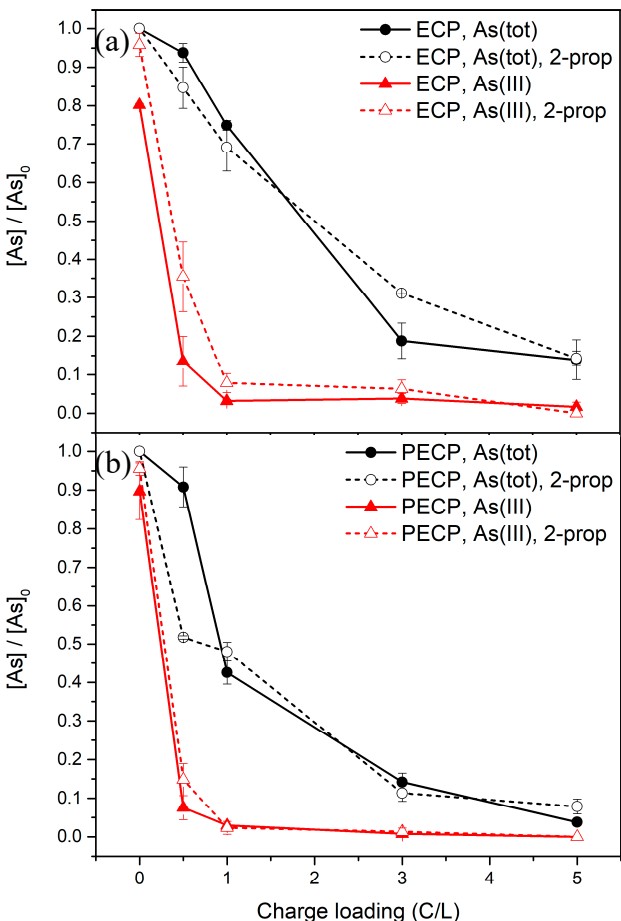

**Figure 4.** Profile of (▲,△) As(III) and (●,○) total arsenic during (**a**) electrochemical peroxidation, (**b**) photo-assisted electrochemical peroxidation. Solid symbols indicate blank experiments in the absence of 2-propanol scavenger. Empty symbols indicate experiments conducted in the presence of 2-propanol scavenger.

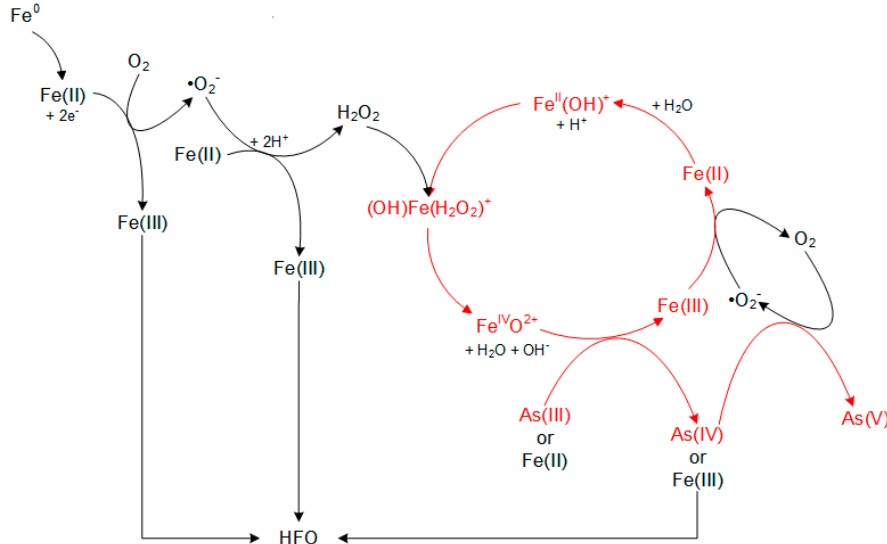

**Figure 5.** As(III) oxidation mechanism in advanced EC processes [45,46]. Highlighted in red is the catalytic reaction involved in the oxidation of As(III) to As(V).

## 2.4. Understanding the Mechanism of Arsenic Oxidation in Advanced Oxidation Processes

In order to fully understand kinetics abatement of As(III) in aqueous samples, a complete insight on kinetic modeling is described herein. Noteworthy is that PECP is a complex hybrid method that combines different possible processes capable of decreasing As(III) concentration over treatment time. Therefore, accounting for the contribution of these different processes, the rate of disappearance can be defined according to Equation (13):

$$\frac{d[As(III)]}{dt} = R_{\bullet OH} + R_{Fe(IV)} + R_{precipitation} + R_{H_2O_2} \tag{13}$$

where $R_{\bullet OH}$ and $R_{Fe(IV)}$ are the rates of As(III) oxidation by •OH and Fe(IV) as the oxidizing agents, respectively. Meanwhile, $R_{precipitation}$ is the rate of precipitation of As(III) by HFO. The term $R_{H_2O_2}$ refers to the chemical oxidation by $H_2O_2$. According to experimental observations, As(III) is barely removed by precipitation, being required its pre-oxidation to As(V) to attain competitive removal [6]. Thus, the contribution of $R_{precipitation}$ can be simplified from Equation (13). Second, direct oxidation of As(III) by $H_2O_2$ in the absence of UV is insignificant to the overall rate [24]. Hence, Equation (13) can be simplified as follows:

$$\frac{d[As(III)]}{dt} = R_{\bullet OH} + R_{Fe(IV)} \tag{14}$$

The simplified model suggests that there are two coexisting reaction mechanisms that may contribute to the abatement of As(III) based on the oxidizing agent involved in the oxidation of As(III). Previous studies proposed a pH-dependent mechanism on the generation of •OH or Fe(IV) as the primary oxidizing agent. It has been identified that •OH acts as the main species in acidic conditions, whereas Fe(IV) is predominant in neutral and alkaline pH [45,46].

The reaction scheme for PECP is outlined in Table 2 and summarized in Figure 5. Here, we first distinguish reactions in acidic and basic solutions. In basic solution, the rate of As(III) oxidation should be given by Equation (15):

$$R^b_{As(III)} = -k_{b4}[Fe(IV)][As(III)] \tag{15}$$

**Table 2.** Relevant reaction scheme involved in advanced EC processes.

| Constants | Reactions | Ref. |
|:---:|:---:|:---:|
| D | $Fe(0) \rightarrow Fe^{2+} + 2e^-$ | - |
| $k_1$ | $Fe^{2+} + O_2 \rightarrow O_2^{\bullet-} + Fe^{3+}$ | [46] |
| $k_2$ | $Fe^{2+} + O_2^{\bullet-} + 2H^+ \rightarrow Fe^{3+} + H_2O_2$ | [46] |
| $k_3$ | $Fe^{2+} + H_2O \rightleftharpoons Fe^{II}(OH)^+ + H^+$ | [46,47] |
| $k_{b1}$ | $Fe^{II}(OH)^+ + H_2O_2 \rightarrow (OH)Fe(H_2O_2)^+$ | [45,47] |
| $k_{b2}$ | $(OH)Fe(H_2O_2)^+ \rightarrow Fe^{IV}O^{2+} + H_2O + OH^-$ | [45,47] |
| $k_{b3}$ | $Fe^{IV}O^{2+} + Fe^{2+} + H_2O \rightarrow 2Fe^{3+} + 2OH^-$ | [46] |
| $k_{b4}$ | $Fe^{IV}O^{2+} + As(III) + H_2O \rightarrow As(IV) + Fe^{3+} + 2OH^-$ | [46] |
| $k_{b5}$ | $Fe^{3+} + O_2^{\bullet-} \rightarrow Fe^{2+} + O_2$ | [46] |
| $k_{b6}$ | $As(IV) + O_2 \rightarrow As(V) + O_2^{\bullet-}$ | [46] |
| $k_{a1}$ | $Fe^{2+} + H_2O_2 \rightarrow Fe(H_2O_2)^{2+}$ | [45,46] |
| $k_{a2}$ | $Fe(H_2O_2)^{2+} \rightarrow Fe^{III}OH^{2+} + \bullet OH$ | [45,46] |
| $k_{a3}$ | $Fe^{2+} + \bullet OH \rightarrow Fe^{3+} + OH^-$ | [46] |
| $k_{a4}$ | $As(III) + \bullet OH \rightarrow As(IV) + OH^-$ | [46] |
| $k_{a5}$ | $As(IV) + O_2 \rightarrow As(V) + O_2^{\bullet-}$ | [46] |
| $\Phi$ | $H_2O_2 \rightarrow 2\bullet OH$ | - |
| $k_{h1}$ | $2HO_2^{\bullet} \rightarrow H_2O_2 + O_2$ | [25,46] |
| $k_{h2}$ | $H_2O_2 + \bullet OH \rightarrow HO_2^{\bullet} + H_2O$ | [25,46] |

Assuming a steady-state approximation for the unstable intermediates, Fe(IV), $(OH)Fe(H_2O_2)^+$, and $Fe^{II}(OH)^+$:

$$R^b_{Fe(IV)} = k_{b2}[(OH)Fe(H_2O_2)^+] - k_{b3}[Fe(IV)][Fe^{2+}] - k_{b4}[Fe(IV)][As(III)] = 0 \qquad (16)$$

$$R^b_{(OH)Fe(H_2O_2)^+} = k_{b1}[Fe^{II}(OH)^+][H_2O_2] - k_{b2}[(OH)Fe(H_2O_2)^+] = 0 \qquad (17)$$

$$R^b_{Fe^{II}(OH)^+} = k_3[Fe^{2+}] - k_{-3}[Fe^{II}(OH)^+][H^+] - k_{b1}[Fe^{II}(OH)^+][H_2O_2] = 0 \qquad (18)$$

From Equation (18),

$$[Fe^{II}(OH)^+] = \frac{k_3[Fe^{2+}]}{k_{-3}[H^+] + k_{b1}[H_2O_2]} \qquad (19)$$

and from Equation (17),

$$k_{b1}[Fe^{II}(OH)^+][H_2O_2] = k_{b2}[(OH)Fe(H_2O_2)^+] \qquad (20)$$

Substituting Equations (19) and (20) to Equation (16), and solving for [Fe(IV)],

$$[Fe(IV)] = \frac{k_3 k_{b1}[Fe^{2+}][H_2O_2]}{(k_{-3}[H^+] + k_{b1}[H_2O_2])(k_{b3}[Fe^{2+}] + k_{b4}[As(III)])} \qquad (21)$$

Substituting Equation (21) to Equation (15), the rate law equation for the oxidation of As(III) in basic solution is

$$R^b_{As(III)} = -\frac{k_3 k_{b1} k_{b4}[Fe^{2+}][H_2O_2][As(III)]}{(k_{-3}[H^+] + k_{b1}[H_2O_2])(k_{b3}[Fe^{2+}] + k_{b4}[As(III)])} \qquad (22)$$

Experimental evidence from the scavenging studies demonstrates the null contribution of •OH. Because the oxidation of As(III) in basic solution proceeds predominantly with Fe(IV) as the oxidizing agent, the rate of As(III) oxidation by Fe(IV) is equal to the rate of As oxidation in basic medium,

$$R_{\text{Fe(IV)}} = R^b_{\text{As(III)}} = -\frac{k_3[\text{Fe}^{2+}]}{\left(1 + \frac{k_{-3}}{k_{b1}}\frac{[\text{H}^+]}{[\text{H}_2\text{O}_2]}\right)\left(1 + \frac{k_{b3}}{k_{b4}}\frac{[\text{Fe}^{2+}]}{[\text{As(III)}]}\right)} \tag{23}$$

In order to account in the model the effect of pH, it should be considered the effect of $[\text{H}^+]$ on the mechanistic relevance of both kinetic components defined in Equation (14). The pH dependency of the rate of oxidation of As(III) by Fe(IV) is described by the term $k_{-3}[\text{H}^+]$ in the denominator. As the solution becomes acidic, $[\text{H}^+]$ increases, and the rate of oxidation decreases. Furthermore, the externally added $\text{H}_2\text{O}_2$ in ECP and PECP increases $[\text{H}_2\text{O}_2]$ in the denominator of Equation (23), which increases the rate of As(III) oxidation. This agrees with our experimental results: that As(III) removal is greater for ECP and PECP.

Indeed, it would be expected that in acidic solutions the rate of As(III) oxidation would be governed by Equation (24):

$$R^a_{\text{As(III)}} = -k_{a4}[\bullet\text{OH}][\text{As(III)}] \tag{24}$$

Likewise, using a steady-state approximation for the unstable intermediates, •OH and $\text{Fe}(\text{H}_2\text{O}_2)^{2+}$:

$$R^a_{\bullet\text{OH}} = k_{a2}[\text{Fe}(\text{H}_2\text{O}_2)^{2+}] - k_{a3}[\text{Fe}^{2+}][\bullet\text{OH}] - k_{a4}[\bullet\text{OH}][\text{As(III)}] + 2\Phi e^a_\lambda - k_{h2}[\text{H}_2\text{O}_2][\bullet\text{OH}] = 0 \tag{25}$$

$$R^a_{\text{Fe}(\text{H}_2\text{O}_2)^+} = k_{a1}[\text{Fe}^{2+}][\text{H}_2\text{O}_2] - k_{a2}[\text{Fe}(\text{H}_2\text{O}_2)^{2+}] = 0 \tag{26}$$

where $\Phi$ is the quantum yield of $\text{H}_2\text{O}_2$ photolysis and $e^a_\lambda$ is the local volumetric rate of photon absorption of $\text{H}_2\text{O}_2$. From Equation (26),

$$k_{a2}[\text{Fe}(\text{H}_2\text{O}_2)^{2+}] = k_{a1}[\text{Fe}^{2+}][\text{H}_2\text{O}_2] \tag{27}$$

Substituting this to Equation (25) and solving for [•OH],

$$[\bullet\text{OH}] = \frac{k_{a1}[\text{Fe}^{2+}][\text{H}_2\text{O}_2] + 2\Phi e^a_\lambda}{k_{a3}[\text{Fe}^{2+}] + k_{a4}[\text{As(III)}] + k_{h2}[\text{H}_2\text{O}_2]} \tag{28}$$

Thus, the rate law equation for the oxidation of As(III) in acidic solution is solved by substituting Equation (28) to Equation (24):

$$R^a_{\text{As(III)}} = -\frac{k_{a4}[\text{As(III)}](k_{a1}[\text{Fe}^{2+}][\text{H}_2\text{O}_2] + 2\Phi e^a_\lambda)}{k_{a3}[\text{Fe}^{2+}] + k_{a4}[\text{As(III)}] + k_{h2}[\text{H}_2\text{O}_2]} \tag{29}$$

Similar to the case above, because the oxidation of As(III) in acidic solution proceeds predominantly with •OH as the oxidizing agent,

$$R_{\bullet\text{OH}} = R^a_{\text{As(III)}} = -\frac{k_{a1}[\text{Fe}^{2+}][\text{H}_2\text{O}_2] + 2\Phi e^a_\lambda}{1 + \frac{k_{a3}[\text{Fe}^{2+}]}{k_{a4}[\text{As(III)}]} + \frac{k_{h2}[\text{H}_2\text{O}_2]}{k_{a4}[\text{As(III)}]}} \tag{30}$$

Finally, Equations (23) and (30) give us the expressions for the rate of competing reactions as described in Equation (14). Note that only the expression for $R_{\bullet\text{OH}}$ has the term for the photolysis of hydrogen peroxide. Thus, we have shown that UV irradiation only affects $R_{\bullet\text{OH}}$, and not $R_{\text{Fe(IV)}}$.

The comparison of the experimental and model simulations of the fraction of As(III) remaining in the solution as a function of charge loading for PECP as described by both mechanisms above

is presented in Figure 6. The model is generated with an initial As(III) concentration of 306 μg/L, initial $H_2O_2$ concentration of 5 mg/L, charge dosage rate of 1 C/L/min, pH of 6.84, and initial dissolved oxygen concentration of 1.08 mg/L. The averaged value of the local volumetric rate of photon absorption is calculated as:

$$e_\lambda^a = S_{reactor} \int_0^x \kappa C I_0 \cdot 10^{-\kappa C x} dx \tag{31}$$

where $S_{reactor}$ is the reactor free surface ($m^2$), $\kappa$ is the spectral-averaged specific absorption coefficient ($m^2$/mol), $C$ is the $H_2O_2$ concentration (M), $I_0$ is the UV irradiance (W/$m^2$), and $x$ is the liquid depth (m) [48].

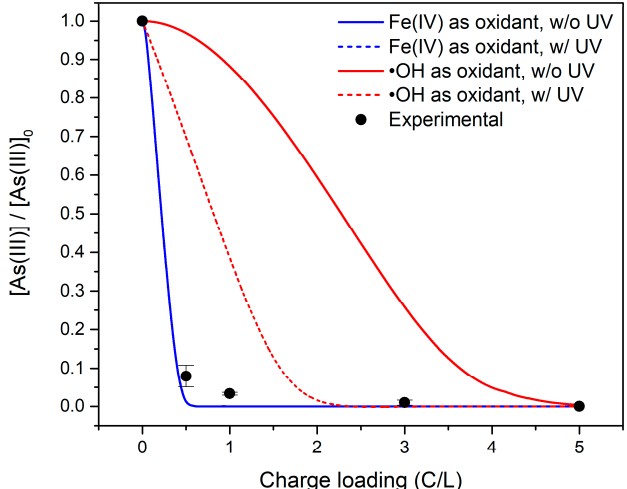

**Figure 6.** Experimental and simulated fractions of As(III) remaining in solution vs. charge loading employing the mechanism for Fe(IV) and •OH as oxidants, with and without UV irradiation. (•) represents experimental data. Solid and dashed lines represent setup without and with UV irradiation, respectively. The blue line represents oxidation by Fe(IV), and the red line represents oxidation by •OH. Note that the blue solid and dashed lines are superimposed. Spectral-averaged specific absorption coefficient, $\kappa$ = 3.89 $m^2$/mol [24]; UV irradiance, $I_0$ = 38.54 W/$m^2$.

The model fits well with the experimental data for the Fe(IV) oxidant regardless of the presence of UV irradiation, as shown by the superimposed blue solid and dashed lines in Figure 6. Furthermore, UV irradiation increases the rate of As(III) oxidation/removal by •OH as oxidant. Therefore, because the dominant mechanism for the oxidation of As(III) is via the Fe(IV) intermediate oxidant, PECP will not give a huge advantage over ECP in the removal of arsenic.

The complete description of this equation model contributes to defining the kinetics of As(III) in a wide range of pH conditions and also incorporates the $H_2O_2$ photolysis effect in the equation for the possible event of simultaneous irradiation. It is important to remark that given the self-buffering capacity of electrocoagulation technologies, the kinetic abatement of As(III) will be predominantly ruled by Fe(IV) oxidation and, therefore, expected to be kinetically defined by Equation (23).

## 3. Materials and Methods

### 3.1. Reagents

Sodium arsenite (NaAsO$_2$ purity ≥ 90%) used as As(III) source was purchased from Sigma-Aldrich (Singapore). Hydrogen peroxide (50% *w/w*) used in advanced EC processes and sodium chloride used as supporting electrolyte were acquired from RTC Laboratory (Quezon City, Philippines) and HiMedia (Mumbai, India), respectively. For •OH scavenging experiments, 2-propanol (≥99.5%) from Duksan Pure Chemicals Co., Ltd. (Ansan, South Korea) was added to the solution. Sodium hydroxide and

hydrochloric acid used to adjust the pH were purchased from QUALIKEMS Fine Chem Pvt. Ltd. (Delhi, India) and RCI Labscan (Bangkok, Thailand), respectively. All solutions were prepared using 18.2 MΩ deionized water.

### 3.2. Experimental Setup and Procedure

The electrochemical reactor (Figure S1) consisted of a 1 L batch reactor with two mild steel electrodes (100 mm × 10 mm × 3 mm). It was operated galvanostatically under monopolar parallel connection using a DC power supply (BK Precision 9111/MCP Lab Electronics M10-TPR3005). The distance between the electrodes was kept constant at 2.0 cm. A digital multimeter (Fluke 117) was used to verify the operating current supplied by the DC power supply. For treatments with UV light exposure, the reactor was enclosed in a black box with two 8W UV-C lamps (Sankyo Denki G8T5) with total irradiance of 3854 μW/cm$^2$ positioned above the reactor [49]. The mixing rate was fixed at 200 rpm using a magnetic stirrer (Corning PC 420D/620D). Prior to electrolysis, electrodes were cleaned by soaking in 1.0 M HCl for 15 min, followed by abrasion using sandpaper. The initial pH was adjusted with 1 M NaOH and 1 M HCl. Solutions were purged with N$_2$ gas prior to treatment to lower the dissolved oxygen (DO) content below 2 mg/L, which represents the DO content of groundwater [50]. Aliquots were collected periodically during the electrochemically-driven treatment and filtered using 0.45 μm syringe filters prior to analyses. The experimental conditions used are summarized in Table S1.

### 3.3. Analytical Methods and Instruments

The solution pH was monitored with a Fisher Scientific (Singapore) accumet AE150 pH meter. As(III) and As(tot) concentrations were measured by anodic stripping voltammetry using Metrohm 946 Portable VA Analyzer. The content of As(V) was determined as the difference between As(tot) and As(III). The oxidation of As(III) in samples taken were immediately quenched with methanol at a 1:1 volumetric ratio [51–53] prior to analysis. The scTRACE gold electrode used in the VA Analyzer was replaced after each experimental run. Arsenic removal was calculated using Equation (32):

$$R_{As}(\%) = \frac{C_o - C_t}{C_o} \times 100\% \tag{32}$$

where $C_o$ and $C_t$ are the initial and final As concentrations, respectively. In addition, As removal capacity per coulomb or mg Fe was computed using Equation (33):

$$RC_{As}(per\ coulomb) = \frac{C_o - C_t}{q} \text{ or } RC_{As}(per\ mg\ Fe) = \frac{C_o - C_t}{m_{Fe}} \tag{33}$$

where $q$ is the charge loading (C/L) applied and $m_{Fe}$ is the mass of electrogenerated iron [27]. The operating cost was calculated by considering the energy ($C_{energy}$, J/m$^3$), electrode ($C_{electrode}$, kg/m$^3$), and hydrogen peroxide consumptions ($C_{H_2O_2}$, kg/m$^3$) as shown:

$$OC = \alpha C_{energy} + \beta C_{electrode} + \gamma C_{H_2O_2} \tag{34}$$

where $\alpha$, $\beta$, and $\gamma$ are costs of electrical energy ($/J), electrode material ($/kg), and hydrogen peroxide ($/kg), respectively. Energy cost considers the consumption due to the electrolytic cell ($C_{EC}$, J/m$^3$) and UV lamp ($C_{lamp}$, J/m$^3$) as shown in Equations (35) to (37):

$$C_{energy} = C_{EC} + C_{lamp} \tag{35}$$

$$C_{EC} = \frac{it_{EC}U}{v} \tag{36}$$

$$C_{lamp} = \frac{P_{lamp}t_{EC}}{v} \tag{37}$$

where $i$ is the applied current (A), $t_{EC}$ is the electrolysis time (s), $U$ is the cell voltage (V), $v$ is the volume of the treated solution (m$^3$), and $P_{lamp}$ is the power consumption of the UV lamps (W). The total electrode consumption was calculated using Equation (38):

$$C_{electrode} = \frac{it_{EC}M_{\text{Fe}}}{nFv} \tag{38}$$

where $M_{\text{Fe}}$ is the molar mass of Fe (55.85 g/mol), $n$ is the number of electrons involved in the oxidation/reduction reaction ($n_{Fe}$ = 2), and $F$ is Faraday's constant (96,485 C/mol).

## 4. Conclusions

Using iron electrodes, both ECP and PECP processes are more effective than EC in reducing the As(III) content of aqueous solutions from ~500 μg/L to below MCL. Enhanced removal is explained by the oxidation of As(III) to As(V) by the in situ generated oxidant species. Experimental evidence demonstrated the null involvement of •OH in the advanced oxidation mechanism at neutral/alkaline pH. In these advanced EC processes, As(III) is first oxidized predominantly by Fe(IV) intermediate at circumneutral pH, and then subsequently removed by adsorption/precipitation by/with HFO. Moreover, UV irradiation does not significantly increase the rate of As(III) oxidation and removal as it is only involved with $H_2O_2$ photolysis to •OH species that is barely involved in the oxidation process at the operating pH conditions. Kinetic modeling proposed accounts for the coexistence of different oxidation processes that are pH dependent. This novel model can provide future insight into technology hybridization and kinetics optimization. Operating costs were estimated as 0.0062 \$/m$^3$ for EC, 0.0060 \$/m$^3$ for ECP, and 0.249 \$/m$^3$ for PECP by considering the energy, electrode, and $H_2O_2$ consumptions only. The large difference in the operating cost of PECP is attributed to the UV lamps used, which greatly increased the energy cost of this process. Hence, ECP is the recommended process for the efficient removal of As(III) in water as a scalable emerging process with high competitiveness.

**Supplementary Materials:** The following are available online at http://www.mdpi.com/2073-4344/10/8/928/s1, Figure S1: Schematic diagram of the electrochemical reactor, Table S1: Summary of experimental conditions.

**Author Contributions:** Conceptualization, M.F.V.M., M.R.C.III and J.D.O.; Methodology, M.F.V.M. and M.R.C.III; Investigation, M.F.V.M.; Resources, J.D.O. and A.C.R.; Writing—Original Draft Preparation, M.F.V.M.; Writing—Review & Editing, S.G.-S., M.R.C.III and J.D.O.; Visualization, M.F.V.M. and S.G.-S.; Supervision, J.D.O.; Funding Acquisition, A.C.R. and J.D.O. All authors have read and agreed to the published version of the manuscript.

**Funding:** This research was funded by the PHIL-ECAR-I project under The Commission on Higher Education—Philippine-California Advanced Research Institutes (CHED-PCARI, IIID-2017-32).

**Acknowledgments:** M.F.V.M. would like to acknowledge the Engineering Research and Development for Technology (ERDT) Program of the Department of Science and Technology—Science Education Institute (DOST-SEI) for the graduate scholarship and research grant. The authors would like to acknowledge Ashok Gadgil of the University of California Berkeley for the discussions. M.F.V.M. and J.D.O. are grateful for the help of Angelica Paz F. Kabigting, Mecaelah S. Palaganas, and Mark Joseph M. Pasciolco in the preliminary experiments.

**Conflicts of Interest:** The authors declare no conflict of interest.

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
