# Peer review of "Arsenic Removal by Advanced Electrocoagulation Processes: The Role of Oxidants Generated and Kinetic Modeling"

_catalysts, doi:10.3390/catal10080928_

Round 1

Reviewer 1 Report

This paper reports on a removal of arsenic using several electrochemical processes such as electrocoagulation (EC), electrochemical peroxidation (ECP), and photo-assisted electrochemical peroxidation (PECP). In this study, especially, the operation cost required for each process and the mechanism of arsenic oxidation were evaluated. The manuscript in general is well done but there were many papers concerning the degradation of arsenic using electrochemical technology. I am not sure what is the novelty of this study. All proposed processes have already been evaluated for arsenic removal. All proposed mechanisms have already been reported. I don’t understand the necessity and meaning of last section “3.4. Understanding the mechanism of arsenic oxidation in advanced oxidation processes”. There was no comparative evaluation between the actual experimental result and the model equation. It was just a formula listing only the description of the existing mechanism. If the authors explain the novelty of this paper so that anyone can accept it, or if they provides fitting data to prove the model equation, I will reconsider.

Author Response

Response to Reviewer #1’s comments.

  • “I am not sure what is the novelty of this study. All proposed processes have already been evaluated for arsenic removal. All proposed mechanisms have already been reported.”

Reply:  We are thankful to the reviewer for his thorough revision and critical input. We agree that we may have not identified clearly the novelty of our study since coagulation treatment to remove arsenic has been previously reported. The novelty of our work relies on the exploration and clarification of the synergistic role of oxidant species electrogenerated in situ by advanced electrocoagulation processes, which is barely explored in literature. Our experiments demonstrate the relevant role that has on removal performance the generation of these oxidants due to the oxidation of As(III) that leads to enhanced removal by coagulation. Furthermore, we explore the role of photo-assisted processes and demonstrate only slight contribution to the overall As removal. The benchmark of these processes not only in terms of performance but also from the economic point of view allow highlighting that even though higher removal is attained with UV-assisted processes the higher cost is detrimental.  The other novelty is the identification of the oxidant species generated and their relationship with the operational pH conditions. This allowed us to develop a complex model that includes different variables on the kinetic removal of As(III). We believe that this work will provide further direction to overcome the arsenic pollution challenge that detrimentally affects developing communities in Asia and South America.

In order to clarify all these points, we have added the following statements in the introduction of our manuscript:

“Previous studies have investigated the oxidation of As(III) by UV/H2O2 [22–26], and the removal of arsenic using EC [6,15,27–31] and ECP [32,33]. Meanwhile, a limited number of papers have explored the PECP process mainly for the treatment of organics via •OH radical formation [34–38]. Elucidation of the role electrogenerated oxidants on the removal of As(III) can allow identifying enhanced removal treatments as well as understanding the fundamental mechanisms of the kinetic process. Therefore, modelling approaches can provide a guideline for future advances to overcome the arsenic pollution challenge that detrimentally affects developing communities.”

  • “I don’t understand the necessity and meaning of last section “3.4. Understanding the mechanism of arsenic oxidation in advanced oxidation processes”. There was no comparative evaluation between the actual experimental result and the model equation. It was just a formula listing only the description of the existing mechanism.”

Reply: This is indeed and excellent question. The scope of Section 3.4. was to develop a comprehensive model capable of describing the role of the different oxidants in function of the pH while incorporating the effect of applied current. This model enables fundamental understanding of the system under different operation conditions and can be used as an optimization tool in the future. Following the reviewer comment and to demonstrate its usefulness we have incorporated a new Figure 6 that shows how the experimental data fits well with the model equation when Fe(IV) is the oxidant, regardless of the presence of UV. Thus, it accurately reflects the behavior of the different systems and their As removal capabilities. 

We are extremely thankful to the reviewer comment that allowed us to develop further our model strengthening its relevance and meaningfulness. Furthermore, we have added the following paragraphs for further clarification:

“The comparison of the experimental and model simulations of fraction of As(III) remaining in the solution as a function of charge loading for PECP as described by both mechanisms above is presented in Figure 6. The model is generated with an initial As(III) concentration of 306 µg/L, initial H2O2 concentration of 5 mg/L, charge dosage rate of 1 C/L/min, pH of 6.84, and initial dissolved oxygen concentration of 1.08 mg/L. The averaged value of local volumetric rate of photon absorption is calculated as per Equation 47.

The model fits well with the experimental data for the Fe(IV) oxidant regardless of the presence of UV irradiation. Furthermore, UV irradiation increases the rate of As(III) oxidation/removal by •OH as oxidant. Therefore, because the dominant mechanism for the oxidation of As(III) is via the Fe(IV) intermediate oxidant, PECP will not give a huge advantage over ECP in the removal of arsenic.

Reviewer 2 Report

The paper discusses the cost comparison of several methods of arsenic removal from water. Based on the data from the literature, it explains the most likely course of reactions leading to the precipitation of arsenic from the solution.
In section 2.3 Analytical methods and instruments, the authors introduce the formulas to calculate the total cost in US dollars per cubic meter of solution. However, the units used in formulas 15 to 19 are incorrectly selected, which causes that each of the members of formula 15 is in different units and their summation is mathematically wrong.
The first term is $ / (1000 * m3 * 60)
The second term is $ * 60 / m3
The third term is $ / m3
For this reason, perhaps, further calculations of the costs of the analyzed processes are burdened with a gross error and the conclusions may be completely wrong.

I strongly recommend using SI units. The kWh is a commonly used unit for the amount of energy, but in equation 15 it causes a mishmash. Use seconds as a measure of time, not minutes.

Author Response

Response to Reviewer #2’s comments.

  • “I strongly recommend using SI units. The kWh is a commonly used unit for the amount of energy, but in equation 15 it causes a mishmash. Use seconds as a measure of time, not minutes.”

Reply:  Upon checking, we have confirmed the use of correct unit conversions for the cost calculations. We apologize for the confusion caused by the units indicated in the paper. We have changed the following units of measure to SI throughout the whole manuscript:

             Line 135: energy (, J/m3)

             Line 137: costs of electrical energy ($/J)

             Line 140: electrolysis time (s)

Round 2

Reviewer 1 Report

I think the authors revised the manuscript accordingly as reviewer commented. The novelty of this paper was explained, and the model fitting data was provided. Thus, I reconsidered it and decided on minor revision.

On Page 4: ‘•OH radical’ is a wrong expression. Radical symbol and terms are used in duplicate.

In Figure 6: The blue dotted line (w/ UV) and the dashed line (w/o UV) overlap so that it cannot be distinguished.

Author Response

I think the authors revised the manuscript accordingly as reviewer commented. The novelty of this paper was explained, and the model fitting data was provided. Thus, I reconsidered it and decided on minor revision.

 On Page 4: ‘•OH radical’ is a wrong expression. Radical symbol and terms are used in duplicate.

Reply: Thank you for this kind suggestion. Now in the revised version, all ‘•OH’ terms in the manuscript have been corrected avoiding the duplication pointed out by the reviewer.

In Figure 6: The blue dotted line (w/ UV) and the dashed line (w/o UV) overlap so that it cannot be distinguished.

Reply: This is an excellent comment. We understand the concern raised by the reviewer regarding the clarity for the reader. We have included this description in the caption of Figure 6 and likewise edited the manuscript to avoid confusion.

Line 327: “Note that the blue solid and dashed lines are superimposed.”

Line 339: “…as shown by the superimposed blue solid and dashed lines in Figure 6.”

Reviewer 2 Report

The authors corrected the inconsistencies in the use of units, but in Equation 15 the second term of the equation has units ($/kg) * (g/m3) while the remaining terms have the resulting units of $/m3. For this reason, to be able to add the whole equation, the second term should be divided by 1000 g/kg. For this reason, in Table 1, Total will change to 0.015, 0.0050 and 0.252, respectively. For this reason, the sentence in lines 177 and 178 "The advanced ECP process shows competitive costs to EC of 0.007 $/m3 with improved performance." becomes untrue. But the comparison of Total (last column of the Table) is also incorrect, because in the EP process, despite the smaller As removal, it consumes 30 minutes compared to 10 minutes for the ECP process. In industrial practice, this means that for EP process larger and more expensive apparata should be built to obtain the required installation capacity.
In my opinion, when comparing costs, authors should use the final As removal as required by the standard.

Equation 13 uses the letter x as the multiplication symbol while Equation 15 through 19 uses the correct "middle dot" sign.
In equation 14, different symbols should be used for removal capacity as they have different units.

Author Response

The authors corrected the inconsistencies in the use of units, but in Equation 15 the second term of the equation has units ($/kg) * (g/m3) while the remaining terms have the resulting units of $/m3. For this reason, to be able to add the whole equation, the second term should be divided by 1000 g/kg. For this reason, in Table 1, Total will change to 0.015, 0.0050 and 0.252, respectively. For this reason, the sentence in lines 177 and 178 "The advanced ECP process shows competitive costs to EC of 0.007 $/m3 with improved performance." becomes untrue.

Reply: Thank you for your keen observation that allowed us to recheck our calculations. We have checked the second term in Equation 15 and we have confirmed that the electrode consumption, Celectrode, has the unit of “kg/m3”. We have already incorporated the “1000g/kg” unit conversion in our calculations when we converted the molar mass of Fe(55.85 g/mol) to “kg/mol” in Equation 19.

But the comparison of Total (last column of the Table) is also incorrect, because in the EP process, despite the smaller As removal, it consumes 30 minutes compared to 10 minutes for the ECP process. In industrial practice, this means that for EP process larger and more expensive apparata should be built to obtain the required installation capacity.

Reply: Thank you for your comment. We agree that the EC process will require a larger and more expensive equipment than ECP process in a continuous large scale set-up. The treatment time is indeed an important factor that must be considered in equipment sizing and capital cost calculations. However, in this study, we were only able to estimate the operating costs from our bench-scale experiments. Thus, we used the “operating cost” in comparing the three processes. Following the reviewer’s observation, we have added the following statement for clarification:

“Moreover, it is to be noted that longer treatment time required by EC does not only affect operational expenditures. Indeed, the required reactor design at larger scales may result in treatment units of higher physical footprint that will also increase capital costs”.

In my opinion, when comparing costs, authors should use the final As removal as required by the standard.

Reply: This is an excellent observation. Calculations were conducted using times close to the attainment of the MCL levels suggested by the WHO. However, we totally agree with the reviewer that a more fair comparison of costs will be obtained using the exact treatment time required to reach the MCL value. Therefore, we have updated the calculations in the manuscript and Table 1 to clearly reflect that. Again, we thank the reviewer for the excellent suggestions and observations that increased the quality of our results’ discussion. See updated table below:

Table 1. Treatment time, arsenic removal capacity, and operating cost required for EC, ECP and PECP processes to reach MCL.

Treatment

tEC, WHO

As Removal Capacity

Operating Cost

min

µg As removed/ Coulomb

µg As removed/ mg Fe

Energy,

$/m3

Electrode,

$/m3

2O2,

$/m3

Total,

$/m3

EC

27.6

16.6

57.3

0.0014

0.0048

-

0.0062

ECP

6.7

63.0

217.8

0.0003

0.0012

0.0045

0.0060

PECP

4.9

111.1

384.0

0.2436

0.0009

0.0045

0.2490

Equation 13 uses the letter x as the multiplication symbol while Equation 15 through 19 uses the correct "middle dot" sign.

Reply: Thank you for your observation. We have already modified Equation 13 following the reviewer’s detailed remark.

In equation 14, different symbols should be used for removal capacity as they have different units.

Reply: We are thankful for the thorough revision of the reviewer. We find that the recommendation is of high relevance and therefore we have updated symbols of Equation 14.
